# Electro-Magnetic and Structural Analysis of Six-Pole Hybrid-Excited Permanent Magnet Motors

Luca Cinti *, Mattia Carlucci *, Nicola Bianchi * and Manuele Bertoluzzo *

Department of Industrial Engineering, University of Padova, 35122 Padova, Italy
* Correspondence: luca.cinti@studenti.unipd.it (L.C.); mattia.carlucci@studenti.unipd.it (M.C.); nicola.bianchi@unipd.it (N.B.); manuele.bertoluzzo@unipd.it (M.B.)

**Abstract:** Potentials and limits of the Hybrid-Excitation Permanent-Magnet (HEPM) synchronous machine are dealt with in this paper. A six-pole machine is taken into consideration, and both parallel and series configurations are analysed and compared. Taking advantage of the rotor excitation coils, the permanent magnet (PM) rotor flux can be adjusted according to the operating speed to improve its performance parameters. The electro-magnetic force is analysed in its first harmonic and in the complete shape. Moreover, a comparison between analytical and numerical formulation has been done for the rotor current control. In particular, the speed range is extended, and electro-mechanical torque and power are increased, as well as the efficiency. It will be shown that the rotor flux reduction by using the excitation winding yields an improvement of the motor performance. The main advantage will be obtained during the flux-weakening operations. In this paper, different rotor topologies will be analysed to highlight the advantages and drawbacks of each of them, and how it is possible to achieve higher speed with higher torque and without high saliency ratio. A magnetic network will be introduced to explain the different contribution of the excitation winding to the rotor flux. Furthermore, a comparison of the amount of the volume of PM, copper and iron in internal permanent magnet (IPM) motor and HEPM motor will be analysed. Actually, an analysis of the harmonic content in the electro-motive force even varying the excitation current and a mechanical stress analysis of each machine will be shown. Finally, it will be verified that the excitation losses represent a minimum component of the total losses.

**Keywords:** finite-element analysis flux-weakening operation; hybrid excitation; interior permanent magnet motor; magnetic analysis; permanent magnet machines; permanent magnet machine control; per-unit system



## 1. Introduction

The growing interest in the electrification of the automotive industry is spurring the development of new efficient electric and electronic components, such as electric vehicle drives [1,2], wireless charging systems [3–5], and converters [6]. In electric vehicle drives, spindle and many industrial applications, the flux-weakening (FW) operations are almost always required. The desired maximum speed is generally between three to five times the base speed. Interior permanent magnet (IPM) synchronous motor drives fed by current-regulated inverter are mainly adopted. Proper motor design and suitable control strategy enable a wide region of operation and extend the application of the synchronous motor drives to several areas formerly reserved to wound-field AC motors or DC motors.

Here in the literature, some papers are available on this subject: Refs. [7,8] investigates some geometries of the IPM motor and the corresponding performance, Refs. [9,10] reports a complete analysis of the IPM motor and [11,12] describes the criteria for designing a synchronous motor, with reluctance and IPM rotor, suitable for flux-weakening applications.

The different direct and quadrature axis flux lines are shown in [13], as well as the corresponding equivalent circuits, and [14] describes the regime operations during flux-weakening of an IPM motor. The studies of [15] highlight the operating limits due to the

rated voltage and current of the inverter and [16] computes the power that can be achieved at various speeds by changing the stator current phase.

The FW operations have been first investigated in [17], showing that the internal voltage can be reduced by the cross coupling between direct and quadrature axis. Later, in [18,19], an analysis using normalized parameters was proposed to consider all the synchronous motor drives.

The effects of saturation are examined in [20] and the iron losses in [21,22]. The synchronous reluctance motor is analysed in [23,24]. Emphasis on the choice of motor parameters of the motor and inverter volt-ampere ratings for variable speed drive applications is given in [25], which reports a design guideline to meet a prefixed torque-speed characteristic. It has been shown that a simplified model is sufficiently accurate to be a useful tool for the design of the motor and for the prediction of its performance in different working conditions [26].

The choice of the PM flux linkage is very important when the IPM synchronous motor is designed. In fact, the higher the PM flux linkage, the higher the rated torque. However, a too high PM flux linkage reduces the motor speed range, that is, the maximum speed that the motor can achieve.

The aim of this paper is to investigate an alternative to the IPM synchronous motor drive. Some alternatives have been studied in the literature [27]. The rotor is modified so that not only PMs are buried in the rotor structure, but also an excitation winding is adopted to regulate the rotor flux. In the resulting structure, the rotor contains both PMs and Rotor Excitation Coils, so that it is referred to as a synchronous Hybrid Excitation Permanent Magnet (HEPM) machine.

The aim of this rotor arrangement is to get the possibility to change the rotor flux during the operation of the motor, so as to achieve a wider speed range. In this paper, different rotor geometries are investigated to get a proper utilization of the inverter power ratings without increasing the motor size.

The HEPM machine combines the advantages of PM machines and wound field machines. In the IPM motor, the PMs produce the total rotor flux, but this flux is constant. In the wound field motor, the flux can be varied but an excitation current is always required. The HEPM motor allows the rotor flux to be reduced or increased by means of a minimum excitation current, which depends on the requirements of the application.

The modulation of the rotor flux, according to the operating speed, yields an increase of the motor performance. In particular:

- the torque increases during the flux-weakening operations,
- the power increases accordingly, and
- the motor efficiency remains high for a wider speed range.

To highlight the benefit of the HEPM machine, some comparisons are reported hereafter referring to six-pole synchronous motor configurations. Series and parallel architecture of HEPM motors are investigated and compared to IPM motor with the same size. In series configurations, the flux produced by the excitation coils passes through the PMs, while in parallel configurations the flux produced by PMs and by excitation coils has different paths [28,29]. Moreover, the fundamental equations applied for HEPM motor are described. The analysis of the steady-state operation of the IPM and HEPM motors exceeding base speed can be readily carried out using the circle diagram theory [30]. It consists in reporting in the $(i_d, i_q)$ plane the constant current, voltage and torque loci, to point out the operation limits and to individuate the most suitable current vector control.

The model is sufficiently accurate for a prediction of the motor performance in different working conditions and suitable for the motor design. The motor performances obtained analytically can be compared with those of a Finite Element (FE) model. The results are generally in satisfactory agreement with the whole operating speed range [31].

In the literature, some studies have been carried out for the comparison between induction motor and IPM motor [32] evaluating the performance of both during the flux-weakening operations. Moreover, a synchronous motor used for Biaxial Excitation Generator for Auto-mobiles [33,34], demonstrating that BEGA has a very large constant power speed range. Recent studies on HEPM motors deal with innovative motors [35], focusing mainly on flux switching motor configurations [36]. New HEPM motor configurations are compared to conventional IPM motors in terms of torque and speed capabilities [37,38].

In such papers, there is not a complete study of the performance of motors with hybrid excitation, that is, considering both PMs and wound field. To fill this gap, hereafter the benefit of adopting a rotor excitation winding that operates together with the PMs is investigated.

The main contributions of this paper are:

- to verify that the rotor flux reduction yields an improvement of the performance of the motor during the flux-weakening operations,
- to verify that with less saliency ratio, it is possible reach high speed with high torque and almost constant power, above the base speed,
- to compare different rotor topologies (series and parallel configuration) in order to highlight the advantages and drawbacks of each one,
- to introduce a magnetic network that is used to explain the different contribution of the excitation winding to the rotor flux,
- to verify that the harmonic content in the electro-motive force is reduced even varying the excitation current, and
- the excitation losses represent a minimum component of the total losses.

In particular, three HEPM motors are compared in terms of FW capabilities, air-gap flux-density distribution and electro-magnetic force. The paper describes the study process carried out to demonstrate the advantages introduced by the hybrid configuration compared to the permanent magnet configuration.

## 2. Analysis and Design of the HEPM/IPM Motor

### 2.1. Equations and Control Adopted in Flux-Weakening Region

The assumptions in the comparison are summarized in Table 1.

As in the IPM motor, torque, voltage and current can be expressed as:

$$T = \frac{3}{2} p \left[ \Lambda_{he} I_q + (L_d - L_q) I_d I_q \right] \tag{1}$$

$$V^2 = \Omega^2 (\Lambda_d^2 + \Lambda_q^2) \tag{2}$$

$$I^2 = I_d^2 + I_q^2 \tag{3}$$

The stator flux linkage components are:

$$\begin{aligned} \Lambda_d &= \Lambda_{he} + L_d I_d \\ \Lambda_q &= L_q I_q \end{aligned} \tag{4}$$

Voltage (2) and current (3) equations represent the voltage and the current limits that affect the machine capability. These equations are written neglecting the resistive component. This assumption is verified if the inductive component of the motors analysed is dominant in comparison to the resistive one. These equations can be plotted in a d−q plane shown in Figure 1: the voltage equation is an ellipse and the current equation is a circle.

**Table 1.** Characteristic shared by all machines.

| Rated working point operations | | | |
|---|---|---|---|
| Stator Rms current density | $J_{stat}$ | 10 | $\mathrm{A\,mm^{-2}}$ |
| Inverter peak current | $I_{peak}$ | 74.4 | A |
| Voltage DC bus | $V_{DC}$ | 600 | V |
| PM characteristics | | | |
| Type | NdFeB | – | – |
| Coercivity | $H_c$ | 850 | $\mathrm{kA\,m^{-1}}$ |
| Relative permeability | $\mu_x = \mu_y$ | 1.049 | – |
| Electrical conductivity | $\sigma$ | 0.667 | $\mathrm{MS\,m^{-1}}$ |
| Stator winding | | | |
| Slot conductors | $n_c$ | 8 | – |
| Machine parallel paths | $n_{pp}$ | 1 | – |
| Rated peak current | $I_N$ | 74.4 | A |
| Stator geometry | | | |
| Air gap | $g$ | 0.89 | mm |
| Number of poles | $2p$ | 6 | – |
| Outer diameter | $D_e$ | 173 | mm |
| Inner diameter | $D_i$ | 115 | mm |
| Axial length | $L_{stk}$ | 150 | mm |
| Slot opening height | $h_{so}$ | 0.5 | mm |
| Wedge height | $h_{wed}$ | 2.75 | mm |
| Slot height | $h_s$ | 17.5 | mm |
| Slot opening width | $w_{so}$ | 2.5 | mm |
| Stator cross section area | $S_{slot}$ | 104.4 | $\mathrm{mm^2}$ |
| Number of Slots | $Q_s$ | 36 | – |
| Shaft diameter | $D_{sh}$ | 38 | mm |

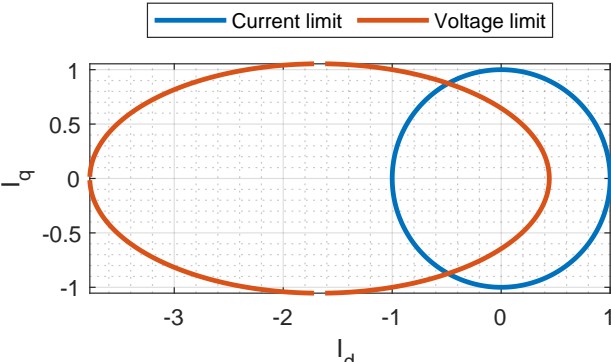

**Figure 1.** Voltage and current equations represented in $I_d/I_q$ plane.

Unlike traditional IPM motor equations, torque (1) and voltage (2) are expressed as a function of:

- $\Lambda_{he}$, which is the sum of PMs flux linkage and Excitation flux linkage,
- p, which are the poles pair,
- $I_d$, which is the direct axis current,
- $I_q$, which is the quadrature axis current, and
- $\Omega$, which is the electrical speed

Therefore, the excitation flux is regulated according to a strategy control, which will be described hereafter and offers an additional degree of freedom compared to typical stator current control.

Once the rated torque ($T = T_N$), the maximum current ($I = I_N$), the maximum voltage ($V = V_N$) and the machine geometry ($L_d$, $L_q$) are selected, it is possible to obtain a formulation of the hybrid flux $\Lambda_{he}$ that maximizes the torque for any speed $\Omega_{fw}$ higher than the rated speed:

$$\Lambda_{he} = \frac{L_q\, L_d(\, I_N\, \Omega_{fw})^2 + V_N^2}{\Omega_{fw}\sqrt{(L_q\, I_N\, \Omega_{fw})^2 + V_N^2}} \tag{5}$$

By manipulating the Equations (1)–(3), the torque has been written as a function of the flux linkage $\Lambda_{he}$. This torque equation was derived with respect to the hybrid flux linkage $\Lambda_{he}$ and the solution is (5). This is the value $\Lambda_{he}$, which maximizes the torque for a given FW speed $\Omega_{fw}$. According to this formulation, the excitation current is modulated to obtain the optimal flux linkage $\Lambda_{he}$ and consequently the stator currents are chosen to satisfy both current and voltage limits. The parameters used to find the flux linkage $\Lambda_{he}$ are evaluated at rated point. The motor parameters are not constant along all the FW regions. The variation of the parameters implies a different behaviour of the machine between the simulation used the analytical formulation and the required performance. The FW performance can be reached with a numerical code that adapts the rotor flux to the variation of the excitation current so as to keep the voltage constant. This is carried out after the stator current and the speed are fixed from the analytical calculation. The evaluation of the voltage is computed with an iterative loop. It should be verified that the difference between its maximum value and the voltage resulting from the simulation is less than 0.5%

### 2.2. Magnetic Network

The machine has been firstly analysed adopting a magnetic network.

Figure 2a,b show the main components of the magnetic network. The excitation coil and the equivalent magnetic lumped parameters are shown in Figure 2a. The excitation coil can be represented by a continuous magnetic voltage source with an amplitude equal to $N_e I_e$, which are the ampere-turns of the excitation coil. The reluctance of air-gap $R_{air}$ is in series with the excitation coil.

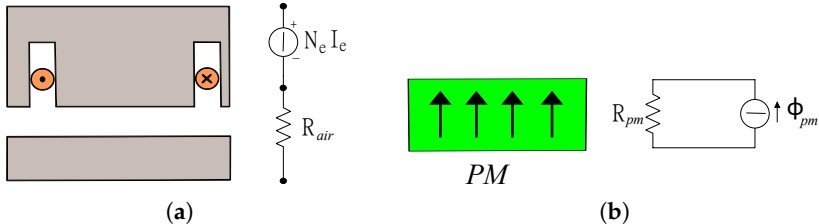

(**a**)                                  (**b**)

**Figure 2.** Elements of magnetic network. (**a**) Excitation winding and magnetic network. (**b**) PM and magnetic network.

Figure 2b represents the PM. The equivalent circuit is a continuous flux source with an amplitude $\phi_{pm}$ in parallel to the PM reluctance $R_{pm}$.

Where the quantity $R_{pm}$ is obtained from:

$$R_{pm} = \frac{t_m}{\mu_r \mu_0 S_{PM}}$$

The parameter $t_m$ is the Pm thickness, $\mu_r$ is the relative differential permeability, $\mu_0$ is the vacuum permeability, $S_{PM}$ the cross-area section of the PM.

The reluctance of air gap $R_{air}$ is:

$$R_{air} = \frac{g}{\mu_0 S_{air}}$$

where $g$ is the thickness of air-gap, $S_{air}$ is the air gap surface.

The PM flux $\phi_{pm}$ is equal to:

$$\phi_{pm} = B_{rem} h_m L_{stk}$$

where $B_{rem}$ is the residual flux density and is obtained from:

$$B_{rem} = \mu_r \mu_0 H_c$$

and $H_c$ is the coercive magnetic field. As hypotheses, all the iron reluctances $R_{fe}$ are neglected because they are not significant with respect to the other quantities.

With these two models, the magnetic network has been built for both IPM and HEPM motors.

Under that theory and assumption, the flux $\Lambda_{he}$ can also be expressed as:

$$\Lambda_{he} = \frac{k_w N}{2} \frac{N_e I_e}{R_{pole}} + \Lambda_{PM} \tag{6}$$

where the reluctance of the pole $R_{pole}$ is the magnetic reluctance corresponding to the rotor excitation winding only, $k_w$ is the winding factor, $N$ is the number of phase stator turns, $N_e I_e$ are the ampere-turns of the excitation coil. These values are different in series and parallel architecture. Series configurations have an $R_{pole}$ higher than the air gap reluctance $R_{air}$. On the contrary, parallel configurations have an $R_{pole}$ close to $R_{air}$.

## 3. SRC-6 Configuration

The series configuration with six rotor coils (SRC-6) is shown in Figure 3a. This machine is compared with a V-shape IPM motor having the same geometry and stator current, with no rotor current. The reluctance of the pole $R_{pole}$ (i.e., the reluctance corresponding to $N_e I_e$ generator only) is mainly due to the PM reluctance $R_{pm}$ (i.e., $R_{pm}/2$ in Figure 3b. Figure 3b shows the corresponding magnetic circuit.

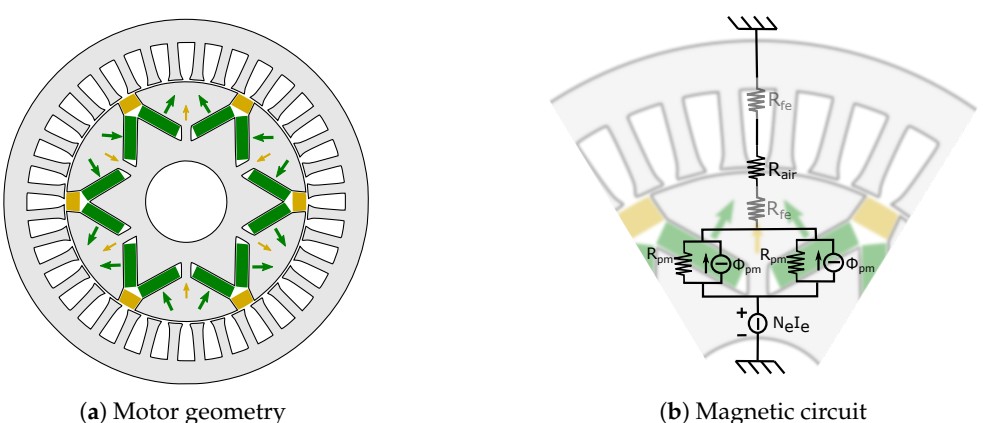

(**a**) Motor geometry　　　　　　　　　　(**b**) Magnetic circuit

**Figure 3.** SRC-6 configuration: geometry and magnetic circuit.

The rotor data geometry is summarized in Table 2.

**Table 2.** IPM and SRC-6 rotor geometry.

| | | | |
|---|---|---|---|
| PM thickness | $t_m$ | 6 | mm |
| PM width | $h_m$ | 21 | mm |
| Rotor slot cross-area section | $S_{slot,exc}$ | 80 | mm$^2$ |
| Rotor current density | $J_{exc}$ | 15 | A mm$^{-2}$ |

### 3.1. Air Gap Flux Density Controlled by the Rotor Excitation Current

The air gap flux contribution due to excitation is reduced because of the high reluctance $R_{pole}$. When the rotor circuit is supplied, the average air gap flux density $B_g$ increases or decreases according to the rotor current direction positive or negative, respectively.

It is shown in Figure 4a. The red curve shows the distribution of $B_g$ with positive excitation current, the dark curve shows the distribution of $B_g$ without any excitation current, and the blue curve is obtained with negative excitation current. The waveform of air gap flux density remains similar to the one with only the PMs in a trapezoidal wave.

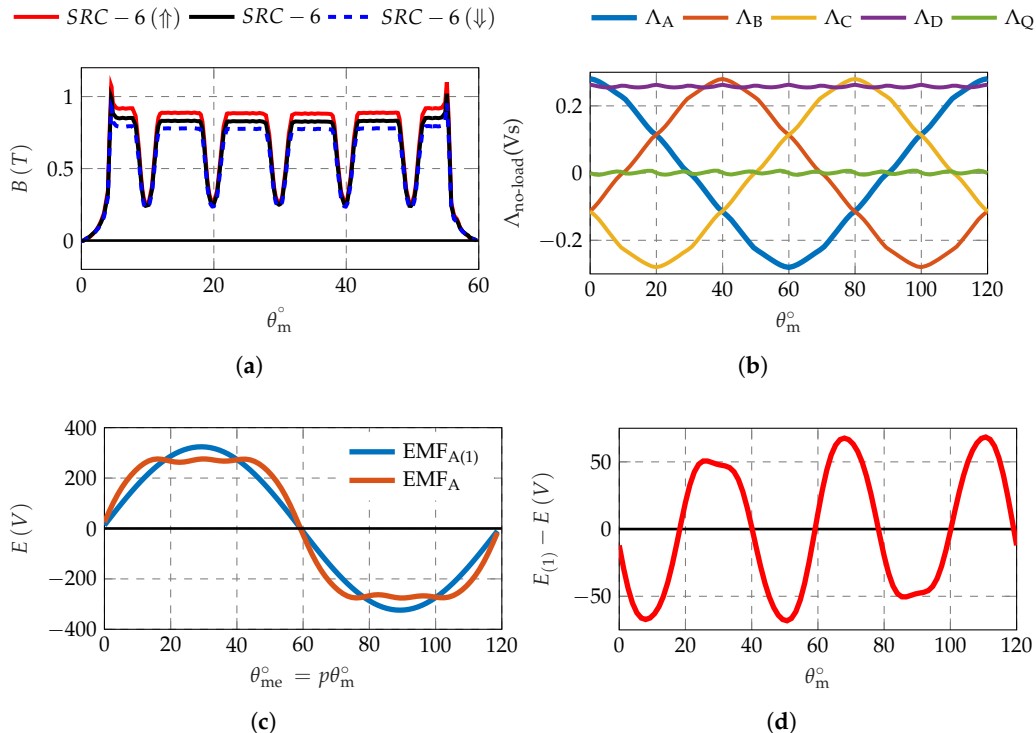

**Figure 4.** Flux density, electro-magnetic force and linkage flux of the SRC-6 machine. (**a**) Air gap flux density distribution versus rotor position. (**b**) No-load stator flux linkage versus rotor position. (**c**) Electro-magnetic force versus rotor position. (**d**) Difference between actual and main EMF harmonic versus rotor position.

### 3.2. Flux Linkage and Electromotive Force

The no-load stator flux linkage, in *abc* and *dq* reference system, produced by this air gap flux density are reported in Figure 4b. Figure 4c reports the behaviour of the *EMF* as a function of the rotor position and the related fundamental component. Therefore, the excitation current does not modify the harmonic content of the air gap flux density.

The electro-motive force can be obtained as:

$$e = -\frac{d\lambda_{he}(t)}{dt} = -\omega\frac{d\lambda_{he}(\theta_m)}{d\theta_m} \tag{7}$$

The no-load flux linkages are not perfectly sinusoidal. They can be expressed by means of Fourier series as:

$$\lambda_{he}(\theta_m) = \sum_{k=1}^{\infty}(a_k \cdot cos(k\theta_m) + b_k \cdot sin(k\theta_m)) \tag{8}$$

Deriving the no-load flux linkage with respect to the time, the various harmonics are multiplied by the corresponding harmonic order $k$. The EMF harmonics result to be amplified with respect to the flux linkage harmonics.

Figure [4]d shows the difference between the actual and the fundamental harmonic of the EMF.

### 3.3. Performance of the SRC-6 Machine

Torque and power curves as a function of rotor speed are shown in Figure [5]a,b. Thanks to rotor flux control, it is possible to reduce the rotor flux on the *d-axis* through both stator currents and rotor excitation current. In the series configuration, due to the high reluctance $\Re_{pole}$, a high variation of the rotor flux is not possible. The SRC-6 motor exhibits a rated torque slightly higher than the IPM motor and an operation range slightly wider than the IPM motor. Figure [5]b shows a maximum SRC-6 motor power of 39 kW at 6000 rpm that drops when increasing the speed.

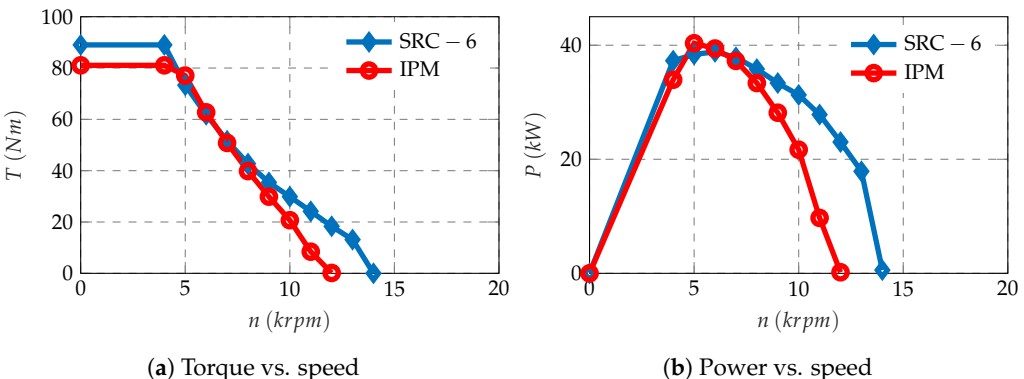

(**a**) Torque vs. speed  (**b**) Power vs. speed

**Figure 5.** Torque and Power of the SRC-6 machine.

Excitation current and Joule losses curves as a function of rotor speed are reported in Figure [6]a,b. The reduction of the air gap flux density is not enough to weaken the rotor flux in a proper way. In other words, the impact of the rotor current is almost negligible in the FW region. Figure [6]a shows a comparison between the excitation current used in the analytical study, assuming the motor working in linear field range, and the excitation current computed with an iterative loop, taking into account the non-linearity of the motor. Figure [6]b shows the component losses that affect the motor from speed 0 to the maximum operating speed. Stator Joule losses $P_{js}$ are plotted with red line, rotor Joule losses $P_{jr}$ with blue line and iron losses $P_{fe}$ with yellow line. In this motor, $P_{jr}$ are not negligible and contribute to the overall losses.

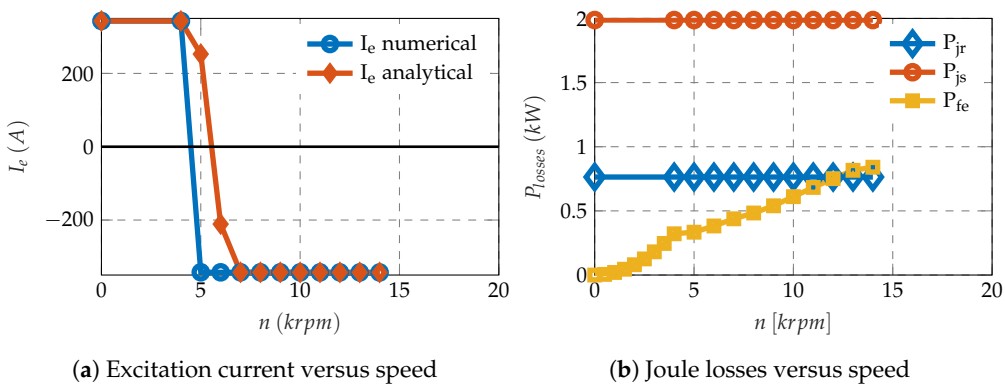

(**a**) Excitation current versus speed  (**b**) Joule losses versus speed

**Figure 6.** SRC-6 machine: Rotor excitation current and joule losses versus rotor speed.

### 4. PRC-6 Configuration

The parallel configuration with six rotor coils (PRC-6) is shown in Figure [7]a. The magnetic circuit is shown in Figure [7]b.

Unlike the SRC-6 machine, the PRC-6 configuration is characterized by a lower $R_{pole}$ because there is an iron path in parallel to the PM path. Therefore, the impact of the rotor

current on the flux variation is significantly higher. Besides, there is a rotor coil for each PM, so that the air gap flux density distribution is equal under each pole.

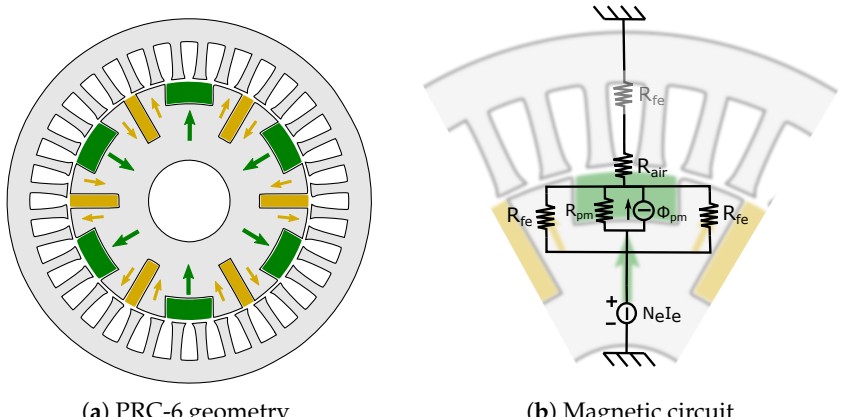

(**a**) PRC-6 geometry　　　　　　　　　(**b**) Magnetic circuit

**Figure 7.** PRC-6 configuration: geometry and magnetic circuit.

The rotor data geometry is summarized in Table 3.

**Table 3.** PRC-6 rotor geometry.

| | | | |
|---|---|---|---|
| PM thickness | $t_m$ | 10 | mm |
| PM width | $h_m$ | 22 | mm |
| Rotor slot cross-area section | $S_{slot,exc}$ | 138 | mm$^2$ |
| Rotor current density | $J_{exc}$ | 10 | A mm$^{-2}$ |

### 4.1. Air Gap Flux Density Modulated by the Rotor Excitation Current

In the PRC-6 machine the contribution of the flux density excitation modifies significantly the flux density distribution in the air gap. The air gap flux density distribution, for PRC-6 configuration, is shown in Figure 8a. The red line represents the HEPM motor with a positive current, the dark line represents the motor with the excitation current equal to zero, while the blue line represents the flux density at the air-gap when the flux due to excitation current is opposite to the PM flux. Figure 8a shows a flux density waveform closer to the sine wave when the current is positive. The contribution of the excitation flux is higher in this configuration than in the SRC-6 motor, due to the low reluctances in the magnetic circuit.

This architecture brings a wider regulation of average air gap flux density. In addiction, compared to SRC configuration, the magnetic voltage drop on the PM is limited, reducing the risk of demagnetizing the PM.

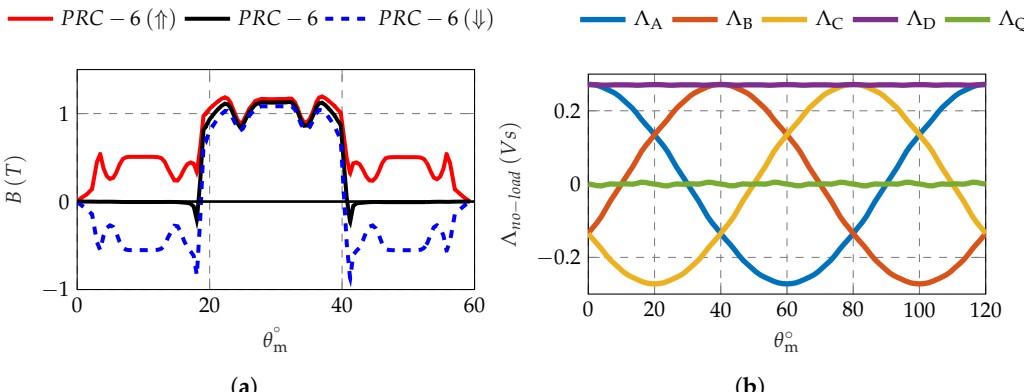

(**a**)　　　　　　　　　　　　　　　　　　(**b**)

**Figure 8.** *Cont.*

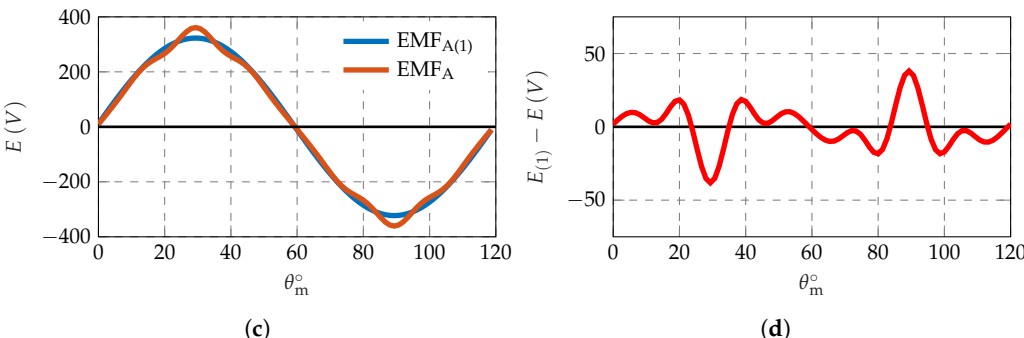

**(c)** **(d)**

**Figure 8.** Flux density, electro magnetic force and flux linkage of PRC-6 machine. (**a**) Air gap flux density distribution versus rotor position. (**b**) No-load stator flux linkage versus rotor position. (**c**) Electro-magnetic force versus rotor position. (**d**) Difference between actual and main EMF harmonic versus rotor position.

### 4.2. Flux Linkage and Electromotive Force

Figure 8b shows the no-load stator flux linkage, they are not perfectly sinusoidal, but compared to the SRC-6 the waves are closer to sinusoidal waves and this yields a reduction of harmonics content. Figure 8c shows the electro-magnetic force EMF behaviour versus positions. With this configuration the harmonic content is quite low.

### 4.3. Performance of the PRC-6 Machine

Figure 9a,b show the torque and the power developed from zero to 20,000 rpm, which is five times the rated speed. At rated point, the maximum torque is similar to that of the SRC-6 motor. However, the reduction of flux allows to achieve a high torque during FW operations. It is worth noticing that once the maximum power is reached, it remains constant in the whole following operation region.

For speeds lower than the rated one, the rotor flux is kept at its maximum value. Above the rated speed, it is possible to regulate the rotor flux so as to maximise the torque for any speed, as indicated in (5).

The computed excitation current is shown in Figure 10a as a function of the rotor speed. The excitation current computed in non-linear simulation results to be different from that computed in the analytical formulation. The high iron saturation is caused by the high current density imposed in the computation, which is typical in motors for automotive application. This saturation impacts on the current behaviour during FW operations. The magnetic circuit is strongly saturated so that the current needed to achieve the proper flux reduction is significantly lower than that analytically computed, as shown in Figure 10a.

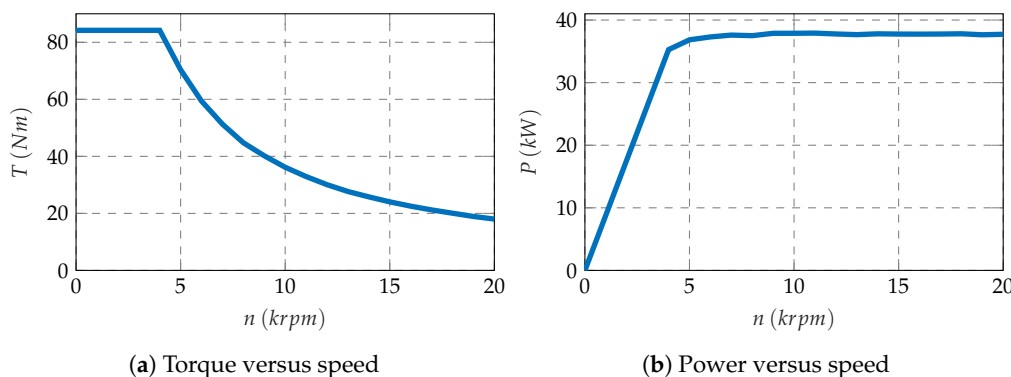

**(a)** Torque versus speed　　　　　　　　　　　　　　　**(b)** Power versus speed

**Figure 9.** Torque and Power of PRC-6 motor.

Differently from the IPM motor, there are additional losses: the rotor joule losses. The corresponding joule losses are reported in Figure 10b together with the stator joule losses

and iron Losses. It is observed that the rotor joule losses are quite low and become almost negligible at high speeds. The rotor current does not reach high values of negative current because the control system regulates the rotor flux to satisfy the voltage and current limits of the inverter.

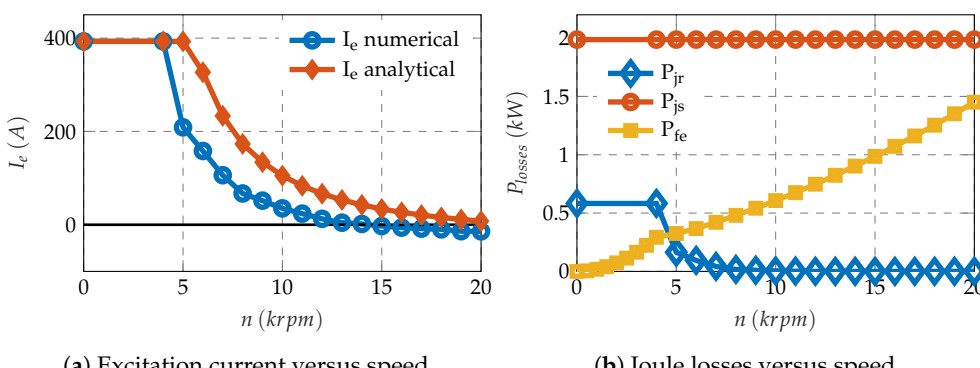

(**a**) Excitation current versus speed　　　　　(**b**) Joule losses versus speed

**Figure 10.** Rotor excitation current and joule losses of PRC-6 motor.

## 5. PRC-2 Motor Configuration

Starting from an IPM motor geometry, this PRC-2 configuration is obtained when two PMs are replaced by two rotor coils, as shown in Figure 11a. The magnetic circuit is shown in Figure 11b.

The rotor data geometry is summarized in Table 4

**Table 4.** PRC-2 rotor geometry.

| | | | |
|---|---|---|---|
| PM thickness | $t_m$ | 10 | mm |
| PM width | $h_m$ | 32 | mm |
| Rotor slot cross-area section | $S_{slot,exc}$ | 206.5 | mm$^2$ |
| Rotor current density | $J_{exc}$ | 10 | A mm$^{-2}$ |

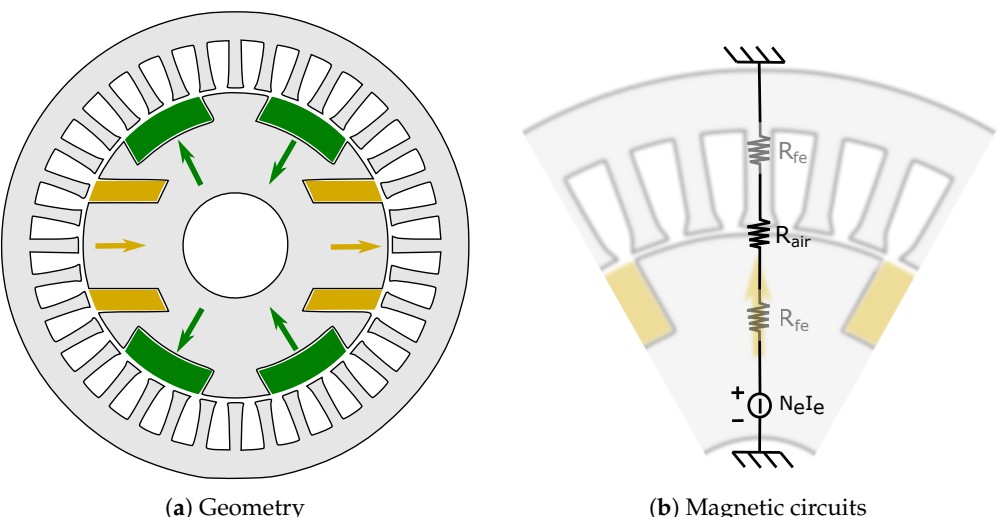

(**a**) Geometry　　　　　　　　　　　　　(**b**) Magnetic circuits

**Figure 11.** PRC-2 configuration: geometry and magnetic circuit.

### 5.1. Air Gap Flux Density Controlled by the Rotor Excitation Current

Figure 12a shows the air-gap flux density in three conditions: red-line refers to the excitation currents producing a flux that replaces a PM flux, dark-line refers to the case with zero excitation currents and blue-line refers to the case with negative current, that is when the excitation flux is opposite to the PM flux. Red-line shows that the excitation current develops a flux with the same average value as the PMs. Blue-line shows that

the average flux is mainly negative on the rotor pole where the coil is located. It's worth noticing that the rotor current produces a flux density variation only in a portion of the air-gap (according to the position of the coils), while the flux remains the same in front of the rotor PM. However, since the stator winding is distributed, a variation of the total flux linkage is obtained.

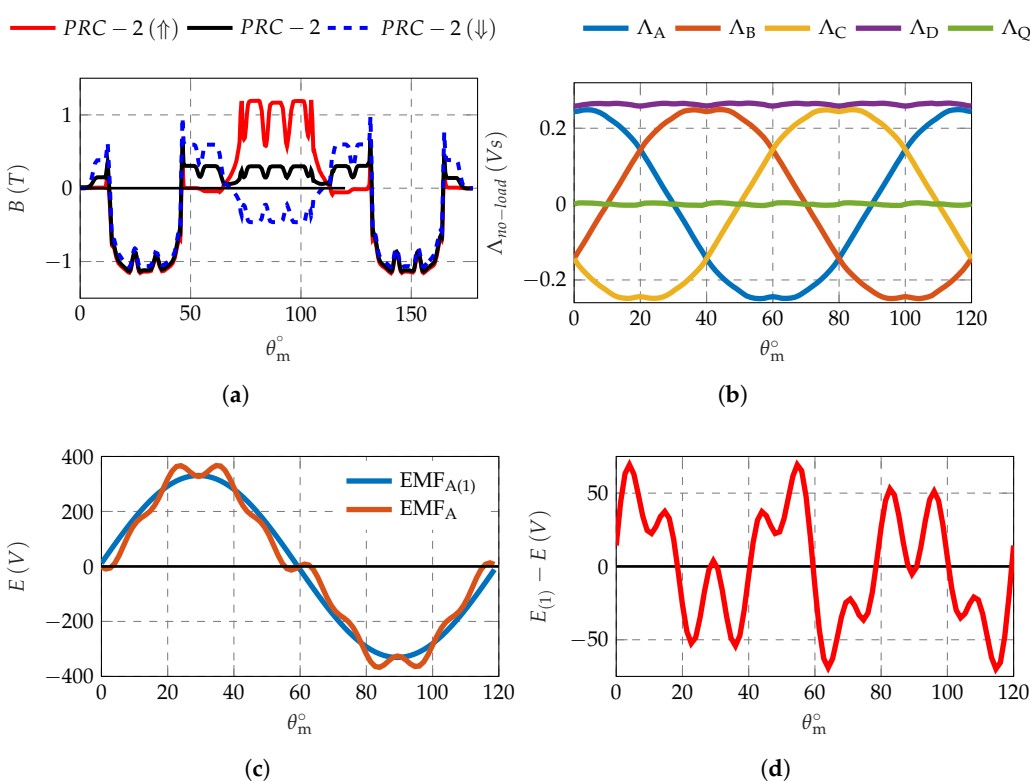

**Figure 12.** Flux density, electro-magnetic force and flux linkage of the PRC-2 machine. (**a**) Air gap flux density distribution versus rotor position. (**b**) No-load stator flux linkage versus rotor position. (**c**) Electro-magnetic force versus rotor position. (**d**) Difference between actual and main EMF harmonic versus rotor position.

### 5.2. Flux Linkage and Electromotive Force

Figure 12b shows the no-load stator flux linkages. They are not perfectly sinusoidal, but similar to the SRC-6, the EMF waveforms are not sinusoidal and there are several EMF harmonics, compared to the PRC-6 machine. Figure 12c shows the electro-magnetic force behaviour as a function of the rotor position at rated speed. The excitation current gives a high contribution substituting the PM, but there is a high harmonic content in the EMF waveforms.

### 5.3. Performance of the PRC-2 Machine

Figure 13a,b show the torque and the power developed from zero to 20 000 rpm, which is five times the rated speed.

Figure 14a shows the behaviour of the excitation rotor current versus speed, which is computed to get the proper rotor flux to maximize the torque at any speed. The result obtained in the non-linear simulation is quite different with respect that achieved by means the analytical formulation. This is due to the iron saturation, which affects the machine performance. This is a consequence of the high current density imposed in the stator.

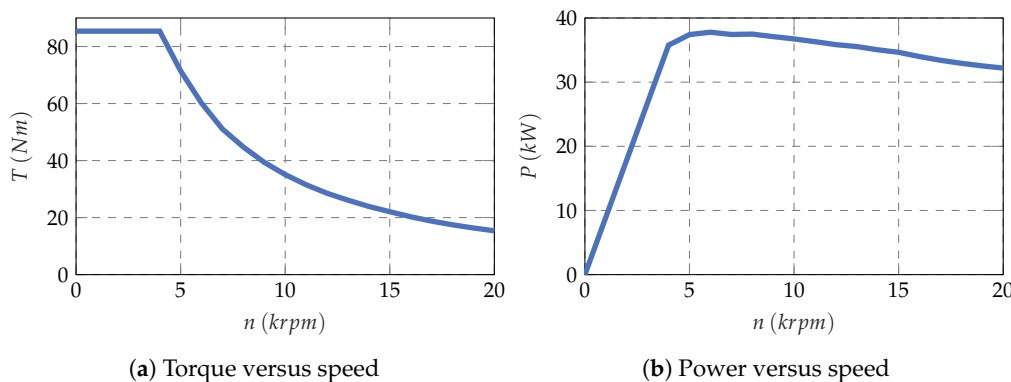

(**a**) Torque versus speed

(**b**) Power versus speed

**Figure 13.** Torque and power versus speed of PRC-2 machine.

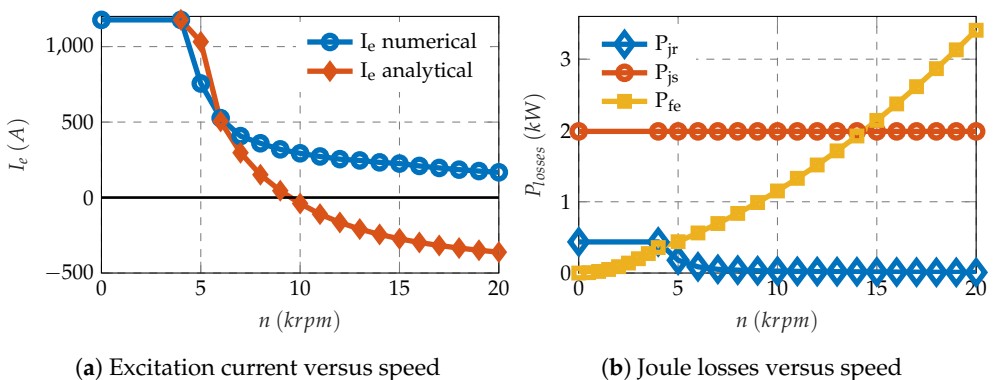

(**a**) Excitation current versus speed

(**b**) Joule losses versus speed

**Figure 14.** Rotor excitation current and joule losses of PRC-2 motor.

## 6. Overall Comparison

### 6.1. Volume and Cost of the Motors Comparison

The overall volume is the same for all from machine to machine, but with different geometry saliency and different amounts of PM. This choice allows us to analyse the performance in the same rated point of each machine. In Figure 15 is shown a comparison of the volume material used in each rotor configuration. The blue bar represents the volume of the iron, the red bar the copper, yellow bar the PM and violet bar the overall volume. The amount of PM is higher in the SRC-2 HEPM and IPM configurations than in PRC-2 and PRC-6. This showed the benefit of the rotor excitation in terms of performance, also limiting the use of rare material. As remarked in the introduction, these benefits bring a reduction of the overall cost of the machine that will not be investigated in detail in this paper.

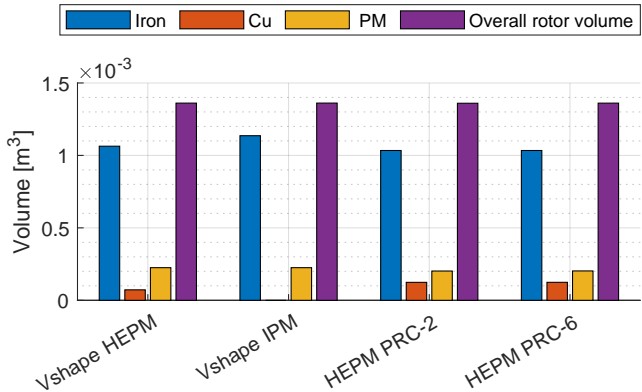

Volume of the configurations analysed

**Figure 15.** Volume of iron, copper, PM and overall of the raw material.

### 6.2. Performance Comparison

Figure 16a,b show a comparison between torque and power curves versus speed developed by the three configurations analysed above. The PRC configurations exhibit a wide speed range compared to the SRC configuration. The torque at rated speed is approximately the same, but at higher speed (e.g., still at two times the rated speed) the torque of the PRC motor is higher than SRC motor.

Figure 16b shows the huge power gap between series and parallel configurations. The overall power developed is higher in PRC motors than SRC motor along all the operating region.

This is due to the different flux density variations when the current produces a flux opposite to the PM flux.

Figure 16c shows the efficiency of the IPM and HEPM motors. At rated speed IPM machine exhibits an efficiency of 93.6% that is higher than any HEPM motor. The maximum at rated speed is reached by the PRC-2 motor with the value of 92.8%. In FW operation the behaviour changes. For speed higher than 7000 rpm the HEPM motor shows a higher overall efficiency. Moreover, the HEPM motors show a higher average value along all the FW operating region.

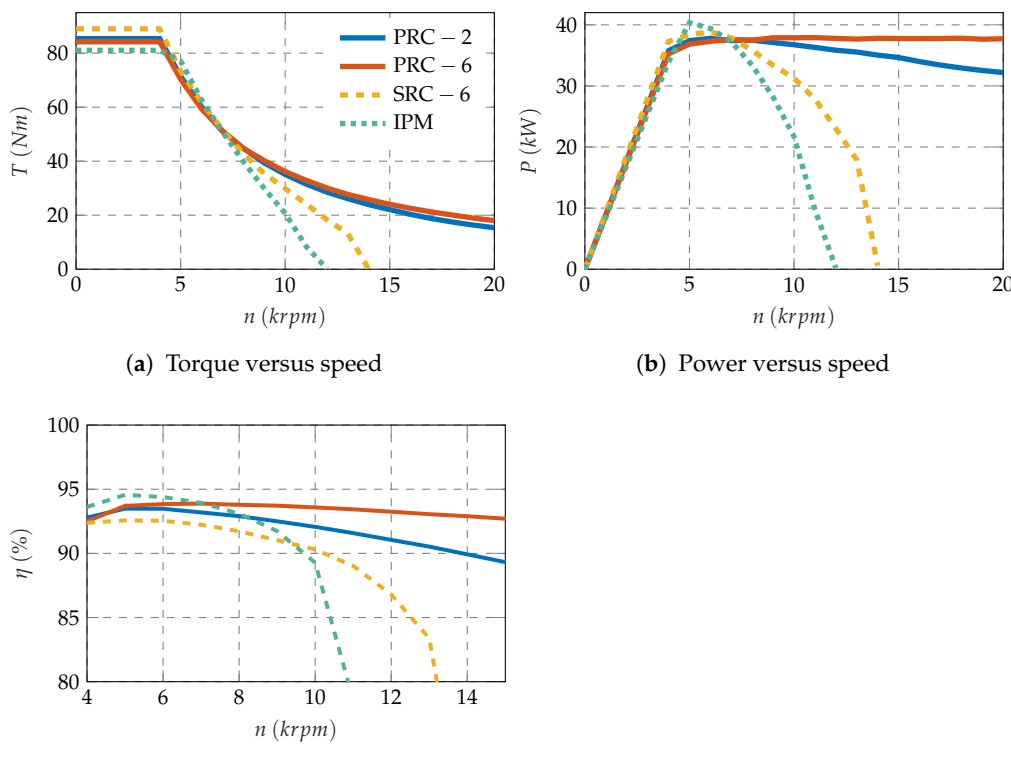

(**a**) Torque versus speed

(**b**) Power versus speed

(**c**) Efficiency versus speed

**Figure 16.** Torque, power and efficiency comparison: PRC-2, PRC-6, SRC-6 HEPM and V-shape IPM motors.

### 6.3. Mechanical Stress

Figure 17a,b show the Von Mises mechanical stress of the PRC-2 and PRC-6 HEPM motors calculated at the maximum speed of 20,000 rpm.

The stress of the PRC-2 is higher than the PRC-6. The symmetry of the PRC-6 brings less centrifugal stress compared to the PRC-2.

Instead Figure 18a,b show the Von Mises mechanical stress of the SRC-6 HEPM and V-shape IPM motors at the speed of 13,000 rpm and 12,000 rpm.

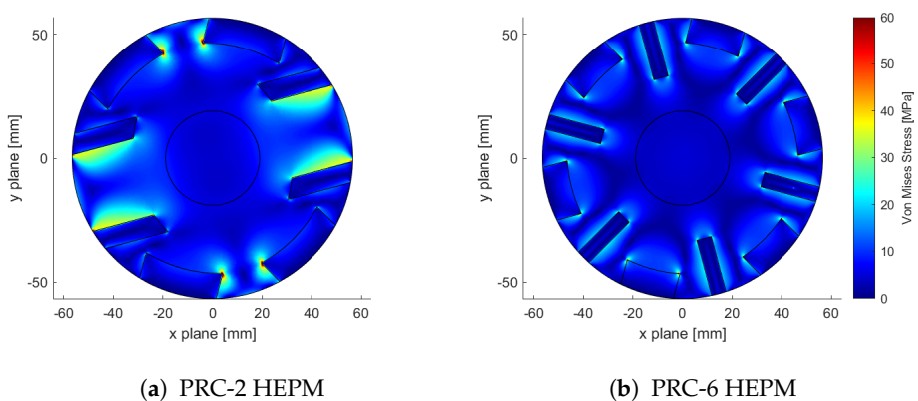

(**a**) PRC-2 HEPM  (**b**) PRC-6 HEPM

**Figure 17.** Mechanical stress of the PRC-2 and PRC-6 HEPM motors.

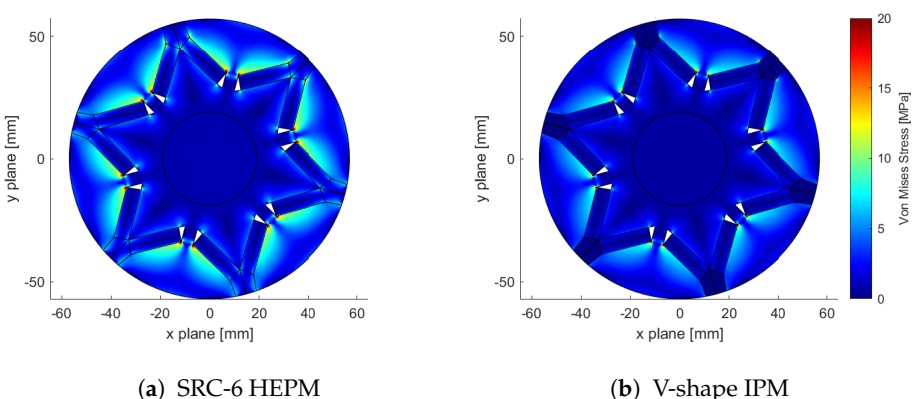

(**a**) SRC-6 HEPM  (**b**) V-shape IPM

**Figure 18.** Mechanical stress of the SRC-6 HEPM and V-shape IPM motors.

In SRC-6 and V-shape IPM motors the speed is lower than PRC-2 and PRC-6 HEPM motors, consequently the centrifugal stress is lower in terms of absolute values. The mechanical stress that affects the PRC-2 HEPM has a maximum value of 58 MPa, PRC-6 motor a value of 30 MPa instead the SRC-6 HEPM and V-shape IPM less than 20 MPa.

For each configuration the maximum level of the stress is much slower than the critical point of the iron crack that is 360 MPa.

The PRC-6 HEPM shows a moderate value of stress compared to the PRC-2 machine at the same speed. Moreover, the distribution of the stress is more equally distributed in the PRC-6 than PRC-2 HEPM motor.

## 7. Conclusions

Three hybrid excitation PM machines are analysed: the first one with series configuration and two other machines with parallel configuration. Advantages and drawbacks are shown for all of them.

The HEPM motor with series configuration results to be not convenient. It exhibits limited performance if compared to the parallel configurations, in terms of both torque and operation speed range. There is a limited rotor flux variation even for a high rotor excitation current. This is because the flux produced by the excitation winding flows through the permanent magnets, which exhibit a high magnetic reluctance.

The PRC-2 HEPM motor exhibits a high rated and flux weakening torque. The variation of the flux is proper to reach a wide speed range. Moreover, the harmonic content is limited. This issue has been verified by a magnetic network according to the position of the excitation coils and the rotor geometry, and a low reluctance is obtained in parallel

to the PM one. However, focusing the reduction of the flux in two poles, the iron losses increase once the speed increases. This is the main drawback of this configuration.

The PRC6 and SRC6 are machines that have lower mechanical stress because the flux reduction is equally distributed along all the 6 poles, instead in the PRC-2 the reduction of flux concerns 2 poles which bring different forces and different vibration modes of the motor.

The PM magnet volume is different from machine to machine, but not the overall volume dimensions. This choice has been done how to analyse in way to have the same rated point performance. Moreover, the HEPM motors have higher FW performance with less use of PM material. Another possible drawback is the possibility of the inverter fault at high speed. This aspect could be negative for the HEPM motors presented. Instead, using the excitation control, it is possible to have the same effect of the high reluctance/magnet torque during the fault, switching off the rotor excitation, reducing the magnet torque and increasing the ratio reluctance/magnet torque.

In conclusion, the PRC-6 HEPM motor exhibits the same advantages of the previous PRC-2 motor, but the PRC-6 motor shows a better behavior in FW operations, a lower stator current reaction and a lower harmonic content in electro-motive force. Actually, by distributing the excitation winding on 6 poles, the iron losses are lower even when the speed increases.

**Author Contributions:** Equal contribution by the all the Authors. All authors have read and agreed to the published version of the manuscript.

**Funding:** This research was funded by Department of Industrial Engineering by means of the grant number BIRD208799, "Hybrid excitation synchronous motor drive for electric vehicles".

**Conflicts of Interest:** The authors declare no conflict of interest.

## Nomenclature

| | |
|---|---|
| $I$ | Current (A) |
| $p$ | Number of pole pairs |
| $I_N$ | Rated motor current (A) |
| $\Lambda_e$ | Flux-linkage due to rotor winding (V s) |
| $I_d$ | $d$-axis current (A) |
| $\Lambda_m$ | Flux-linkage due to PMs (V s) |
| $I_q$ | $q$-axis current (A) |
| $\Lambda_{he}$ | Total hybrid flux-linkage (V s) |
| $L_d$ | $d$-axis inductance (H) |
| $\Omega$ | Electrical speed (rad s$^{-1}$) |
| $L_q$ | $q$-axis inductance (H) |
| $\Omega_{max}$ | Maximum electrical speed (rad s$^{-1}$) |
| $T_{fw}$ | Flux weakening torque (N m) |
| $\Omega_{fw}$ | Flux weakening electrical speed (rad s$^{-1}$) |
| $T$ | Electromagnetic torque (N m) |
| $V$ | Motor phase voltage (V) |
| $V_N$ | Rated motor voltage |
| MTPV | Maximum Torque Per Volts |
| MTPA | Maximum Torque Per Amps |
| PM | Permanent Magnet |
| FW | Flux-weakening |
| HEPM | Hybrid Excitation Permanent Magnet |
| MTPV | Maximum Torque Per Volts |

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
