# Peer review of "Electro-Magnetic and Structural Analysis of Six-Pole Hybrid-Excited Permanent Magnet Motors"

_electronics, doi:10.3390/electronics10172051_

Round 1

Reviewer 1 Report

The Authors mentioned about adopting a rotor excitation winding that operates together with the PMs investigation. in this article, they explained very detailed explanations of HEPM motors and their comparison done in terms of FW capabilities, air-gap flux-density distribution and electro-magnetic force and also mentioned  their advantages. I think this paper is suitable for Electronics journal. I am here accepting this manuscript for publication.

Reviewer 2 Report

The authors have touched upon an actual topic where a permanent magnet motor is analyzed. This is a hybrid technology, where many, as reflected in the review, are looking for a way to maximize performance and minimize heating. The authors have developed an innovative engine, where its data are given, its structure and equivalent circuit are shown. Simulation results are shown, which are compared with known structures. I would like to separately thank the authors for the competent structure of the article, which is not oversaturated with text or figures, it is interesting to read it. A small comment, the authors provide a list of sources used, and among them only 11 out of 31, over the past 5 years. There is not so much modern literature, I would like to see more relevant literature, without begging, nevertheless, the literature of the 80s and 90s, which is undoubtedly relevant in our time. Is it possible to increase the literature from modern sources, at least more than 50% of all sources.

Reviewer 3 Report

This manuscript reports "Electro-magnetic and structural analysis of six-pole
Hybrid-Excited Permanent Magnet Motors".  The work is interesting. It is suggested to be accepted for publication after including the below corrections with major revision.

The language of the manuscript should be elaborated.

Several important references should be cited in the revised manuscript.

The quality of the images and figures should be improved.

The complete terms of all the abbreviations should be mentioned before their first used in both abstract and main content of the revised manuscript.

Reviewer 4 Report

Actually in this paper authors have discussed various parameters like torque, power etc. with respect to speed. Whereas in the performance comparison section authors compared the performances of various materials like cu, iron etc. So please add the performance comparison of various parameters for each material so that it could be understandable. 

Round 2

Reviewer 3 Report

The manuscript is revised properly. It can be accepted for publication after approve of the editor.